# Learning Riemannian Metrics for Interpolating Animations

## Abstract

We propose a new family of geodesic interpolation techniques to perform upsampling of low frame rate animations to high frame rates. This approach has important applications for: (i) creative design, as it provides a diversity of interpolation methods for digital animators; and (ii) compression, as an original high frame rate animation can be recovered with high accuracy from its subsampled version. Specifically, we upsample low frame rate animations by interpolating the rotations of an animated character's bones along geodesics in the Lie group $SO(3)$ for different invariant Riemannian metrics. For compression, we propose an optimization technique that selects the Riemannian metric whose geodesic most faithfully represent the original animation. We demonstrate the advantages of our approach compared to existing interpolation techniques in digital animation.

## 1 Introduction

**Upsampling for creative design** When digital animators create animations, they must position their character's limbs in 3D space—called a *pose*—to make the character perform the actions they want. Animators do so using a *rig*, which is a skeleton made of connected bones in a tree structure, whose positions and rotations control the character mesh that it is attached to. However, the labor of posing $k$ characters with $b$ bones at $m$ frames grows quickly as $\mathcal{O}(k * b * m)$ and in fact becomes impractical at 30 frames per second.

Instead, animators block out key poses in time—called *keyframes*. A keyframe contains the location, rotation, and scale of a bone at a specific time. After posing their character at these keyframes only, animators rely on automatic *in-betweening* to create an *animation curve*, that is: a complete animation of the character through time. In-betweening is a cornerstone of the creation process, as it automatically generates poses in between two consecutive keyframes: it effectively performs interpolation to find intermediate poses. In-betweening can use different interpolation schemes and types of artistic controls available in animation software.

Often, when automatic in-betweening is not satisfactory, animators still need to edit the animation curves directly, which means manually correcting the automatically generated poses to fit their artistic intention. Current industry-standard interpolation schemes such as spherical linear interpolation (SLERP) fail at interpolating very sparse keyframes, so animators use more keyframes to achieve higher fidelity in-betweening thus increasing memory storage and computation cost. Alternatively, state-or-the-art deep learning methods such as Oreshkin et al. (2023) can handle sparse key-frame interpolation, but rely on having hours of previous animation data for interpolations, and do not generalize to different rig models.

Consequently, there is an interest in providing novel interpolation methods to increase the diversity of automatically generated animation curves, enhancing and speeding up the creation process of digital animators while minimizing the number of keyframes required for high fidelity in-betweening.

**Upsampling for compression** Beyond animation creation for digital arts, upsampling techniques are crucial for animation compression. Storing every location, rotation, and scale ($k$) of every bone ($b$) in the rig of a character for every frame ($m$) of an animation can impose large memory requirements ($O(k*b*m)$). Being able to compress an animation into a handful of keyframes is of practical importance. From the perspective of compression, upsampling techniques aim to faithfully recover



GT     PC     LC     Slerp     Ours

Figure 1: Comparison of traditional interpolation techniques with the proposed geodesic interpolations. From left to right: ground truth animation, piecewise constant, linear (cartesian), spherical linear (slerp), and our geodesic interpolation. In this frame our geodesic interpolation most closely matches the original.

an original animation from its compressed version composed of only a subset of poses. Upsampling beyond the original frame rate allows animators to then create an animation curve with an arbitrarily high frame rate. Thus, there is a motivation for researching new upsampling techniques that can achieve high accuracy in recovering original animations and even enhance them by increasing their original frame rate.

**Contributions** We propose a new family of interpolation techniques—called geodesic interpolations—to perform upsampling of low frame rate animations to high frame rates for creative expression and compression in digital animation. Geodesic interpolation generates characters' poses in between two keyframes by computing a geodesic curve linking these keyframes in the space of poses. The space of poses is represented as the set of rotations defining the orientations of the character's bones: *i.e.*, as a Lie group $SO(3)^B$ where $B$ is the number of bones. As geodesic curves depend on the choice of Riemannian metric on $SO(3)^B$, we propose a family of geodesic interpolation techniques parameterized by a family of Riemannian metrics on $SO(3)^B$.

From the perspective of creation, the diversity of Riemannian metrics yields a novel diversity of in-betweening methods for digital creators. From the perspective of compression, we propose to rely on a gradient-free optimization technique to find the geodesic that best interpolates, and thus compresses, a given original animation. Our interpolation schemes applied to animations subsampled at a very low frame rate achieves high accuracy reconstruction of original high frame rate animations. Our work thus also allows for extreme downsampling of an animation while being able to faithfully upsample it back up to a high resolution animation.

Specifically, our contributions are as follows:

1. We propose a new family of geodesic interpolations for 3D animations parameterized by Riemannian metrics on the space of poses.

2. We explore how these geodesics characterize motion, providing guidelines for digital animators that can rely on a new diversity of interpolated motions for creation.

3. We propose a gradient-free optimization technique to select the Riemannian metric whose geodesic most closely matches an original animation curve, and subsequently achieve a compression of animations that outperforms traditional techniques. To our knowledge, this is the first time that Riemannian metric learning is applied to computer graphics.

## 2    RELATED WORKS

**Interpolation** Animation software traditionally relies on different types of interpolation techniques Haarbach et al. (2018). *constant interpolation*, which corresponds to simply holding the character rig still in between frames. Shown in 2, constant interpolation is jittery and tends to have the "stop motion effect" where the character's motion looks choppy. The second type, *linear interpolation*, is easy to compute, but can yield unrealistic motions due to the non-linear nature of human movement. Linearly interpolating the orientations of the bones represented as rotation matrices or quaternions will fail since neither the rotations matrices nor the quaternions are linear: for example, the average of two rotation matrices, or two quaternions, is not necessarily a valid rotation matrix or quaternion. Consequently, specific techniques are introduced to address the nonlinearity of the space of rotations Dam et al. (1998), *e.g.*, making sure that the quaternions lie on the unit sphere through spherical linear interpolation (or slerp) Shoemake (1985). The third type of interpolation, *spline interpolation*, uses a set of piecewise polynomial interpolating functions making a cubic Bezier splines Bézier (1972) (see inset). Applying cubic spline interpolation to rotations again requires

adapting the geometry of the rotation space through spherical spline quaternion (squad) Shoemake (1987). Other interpolation methods exist Mukai & Kuriyama (2005), but are seldom implemented in animation software. Newer machine learning methods such as Oreshkin et al. (2023); Harvey et al. (2020); Zhang & van de Panne (2018) rely on large datasets with hours of animation video for training their interpolation models. In contrast to widely used methods such as slerp, deep learning techniques can handle much larger gaps between keyframes, however, their interpolation functions do not generalize to new rigs. Additionally, Zhang & van de Panne (2018) exhibits loss of motion continuity near keyframes, and Oreshkin et al. (2023) can only interpolate on rigs seen during training with transitions from the training dataset.

Our interpolation technique is most closely related to linear interpolation and slerp. Our method works on arbitrary rigs with sparsely keyframed samples without the use of any training data while providing a novel class of interpolation schemes beyond slerp's by using non-spherical manifolds. In contrast with spline interpolation, which generalizes the interpolating curves from lines to splines, we keep linear curves (geodesics) but change the geometry of the space in which these curves live: changing the very definition of "linear".

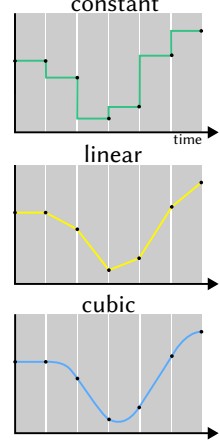

Figure 2

**Artistic control for creative design** Animation curves are the standard representation of animation data. Expert artists edit these curves directly to tweak their animation to look the way they intend. Intuitive alternatives for real time control such as motion matching, proposed by Büttner & Clavet (2015) provide a data-dependent way to create interpolated motions. While standard motion matching methods use high amounts of memory and scale linearly in computation Holden et al. (2020) solves these issues using a neural approach. Both standard and neural motion matching, however, take large amounts of data in order to creatively interpolate trajectories. Other methods are less data-dependent, but require skill and time from users. Staggered poses lets animators refine motion at slightly different times than the keyframes Coleman et al. (2008). Ciccone et al. (2019) create a simplified control that allows for constrained editing of the tangents of animation curves without adding new keyframes. Similarly, Koyama & Goto (2018) offer animators a way to change the tangents of animation curves using sliders. Other work focuses on the problem of providing the keyframes themselves in more user-friendly options such as sketch-based methods where artists can specify keyframes with 2D drawings or sketch motions with short strokes Davis et al. (2006); Guay et al. (2015).All of these approaches still require heavy manual labor from animators. Additionally, they require skilled users with a deep understanding of industry-standard animation software. Given the need for simple, automated techniques, researchers have also recently leveraged neural networks to generate in-between poses Harvey et al. (2020); Zhang & van de Panne (2018). Yet, these approaches are more computationally intensive than traditional interpolation techniques, and less intuitive for artistic control due to the high number of hyperparameters affecting the end results.

By contrast, our geodesic approach is automated with few parameters which also have intuitive kinematic meaning which make them suitable for use by novice animators.We provide an accessible, yet intuitive, alternative for artistic control during the creation process.

**Animation compression** A large animation sequence can be compressed into a small set of frames to save space. During decompression, these frames are interpolated to accurately recover the original animation. Principal Component Analysis is a common compression technique used by Alexa & Müller (2000); Váša et al. (2014) on mesh animations (not rigs) in order to reduce storage. Data-driven techniques analyze patterns in motion capture data Gu et al. (2009) or use a weighted combination of a few basis meshes Abdrashitov et al. (2019); Le & Deng (2013). In contract, our method uses no prior data and works on rig-based animations. We **learn** a Riemannian metric which allows us to pick keyframes that compress an animation sequence in a way that allows us to decompress with the highest accuracy.

**Models of Articulated Motions and Rotations** Outside of digital animation, other works have proposed modelling articulated motions by leveraging Riemannian geometry on manifolds. In robotics, Žefran & Kumar (1998) leverages the variety of Riemannian metrics on the space of rigid body motions $SE(3)$ to provide new kinematics insights. In motion capture technology, Tiwari

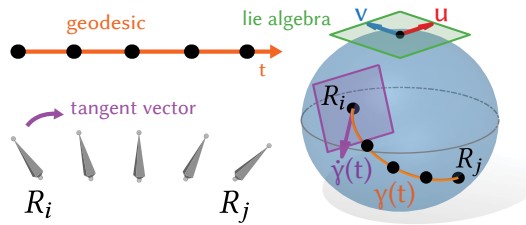


(a) For ease of explanation, we represent $SO(3)$ as a sphere. At the identity of the group, we define an inner-product for all vectors $u, v$ in the tangent space (green). The vector $\dot{\gamma}(t)$ in the tangent space at $R_i$ (purple) is the velocity of the parameterized curve going to $R_j$. The geodesic curve $\gamma(t)$ (orange) is the shortest path between two rotations $R_i$ and $R_j$. In this example we interpolate 3 in-between rotations along the geodesic (black dots).

(b) Changing $\alpha, \beta$ values can be thought of as deforming the group. The distance between the same 2 rotations changes. Our Pose Lie group is a product of manifolds, one for each bone.

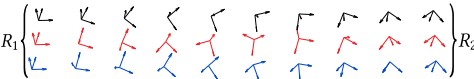

(c) We verify experimentally that updating $\alpha, \beta$ leads to different geodesics, and thus different trajectories despite the same start and end states.

Figure 3: Explaining how a geodesic on a manifold can interpolate trajectories. 397.48499pt

et al. (2022) propose to generate new human poses from high dimensional domain of quaternions in order to enhance performances on downstream tasks such as pose denoising or pose recovery. Authors from Guo et al. (2020) leverage Lie Algebra theory to represent natural human motions and increase capabilities of human poses conditional generation from action types. In Zhou et al. (2016), authors propose to integrate geometric constraints kinematic into a deep learning model for general articulated object pose estimation.

Beyond dedicated application domains, research in applied mathematics has also investigated theoretical properties and possible extensions of interpolation specifically designed for $SO(3)$ with the fixed canonical Riemannian metric Park & Ravani (1997), or for general Riemannian manifolds Gousenbourger et al. (2016) including data fitting schemes Bergmann & Gousenbourger (2018); Gousenbourger et al. (2019). However, the canonical metric is not the only geometry that can equip the non-linear space of rotations $SO(3)$. The work by Huynh (2009) presents a comparison and theoretical analysis of metrics and distances for 3D rotations. Yet, none of these application-driven or theoretical research considers learning the optimal Riemannian metric of $SO(3)$—which is what we propose to do here.

# 3 BACKGROUND

We introduce elements of Riemannian geometry and Lie groups that support our approach.

## 3.1 MANIFOLDS AND RIEMANNIAN METRICS

Lie theory and Riemannian geometry provide mathematics to precisely define the poses of animation characters, specifically the rotation of each joints of a character, and to design novel interpolation techniques. We refer the reader to Guigui et al. (2022) for mathematical details. We will represent the space of possible animated character poses as a Lie group equipped with a Riemannian metric. We define these concepts here.

**Definition 3.1 (Riemannian metric)** *Let $\mathcal{M}$ be a $d$-dimensional smooth connected manifold and $T_p\mathcal{M}$ be its tangent space at point $p \in \mathcal{M}$. A Riemannian metric $<,>$ on $M$ is a collection of inner products $<,>_p: T_p\mathcal{M} \times T_p\mathcal{M} \to \mathbb{R}$ on each tangent space $T_p\mathcal{M}$ that vary smoothly with $p$. A manifold $\mathcal{M}$ equipped with a Riemannian metric $<,>$ is called a Riemannian manifold.*

A Riemannian metric $<,>$ provides a notion of *geodesic distance* `dist` on $\mathcal{M}$. Let $\gamma : [0, 1] \to \mathcal{M}$ be a smooth parameterized curve on $\mathcal{M}$ with velocity vector at $t \in [0, 1]$ denoted as $\dot{\gamma}_t \in T_{\gamma(t)}\mathcal{M}$. The length of $\gamma$ is defined as $L_\gamma = \int_0^1 \sqrt{<\dot{\gamma}_t, \dot{\gamma}_t >_{\gamma_t}} dt$ and the distance between any two points $p, q \in \mathcal{M}$ is: $\text{dist}(p, q) = \inf_{\gamma:\gamma(0)=p,\gamma(1)=q} L_\gamma$. The Riemannian metric also provides a notion of geodesic.

**Definition 3.2 (Geodesic)** *A geodesic between two points $p, q$ is defined as a curve which minimizes the energy functional:*

$$E(\gamma) = \frac{1}{2} \int_0^1 <\dot{\gamma}(t), \dot{\gamma}(t)) >_{\gamma(t)} dt. \tag{1}$$

*Curves minimizing the energy $E$ also minimize the length $L$: geodesics are locally distance-minimizing paths on the manifold $\mathcal{M}$.*

Intuitively, a geodesic is the generalization of the straight lines from vector spaces to manifolds, see Figure 3a. We note that the notion of geodesic depends on the notion of geodesic distance, and thus on the choice of Riemannian metric on the manifold $\mathcal{M}$. Different Riemannian metrics yield different geodesics between two given points.

### 3.2 LIE GROUPS AND METRICS

In the context of animation interpolations, we will consider specific manifolds: Lie groups.

**Definition 3.3 (Lie group)** *A Lie group is a group $(G, \cdot)$ such that $G$ is also a finite dimensional smooth manifold, and the group and differential structures are compatible, in the sense that the group law $\cdot$ and the inverse map $g \mapsto g^{-1}$ are smooth. Let $e$ denote the neutral element, or identity of $G$. The tangent space $T_eG$ of a Lie group at the identity element $e \in G$ is called the Lie algebra of $G$.*

The set of all 3D rotations forms a Lie group. This group is referred to as $SO(3)$, the special orthogonal group in three dimensions. It is defined as: $SO(3) = \{R \in M_3(\mathbb{R}) | R^T R = I_3, \det(R) = 1\}$, where each element is a 3D rotation matrix $R$. Its Lie algebra is denoted by $so(3)$. The Lie algebra is a vector space of dimension 3, which is also called the dimension of the Lie group $SO(3)$. Consider an animation showing a character with $B$ bones. Each bone in the skeleton, or rig, is associated with a joint that has some 3D orientation, which we represent as a rotation matrix $R \in SO(3)$.

**Definition 3.4 (Pose Lie group)** *Consider a character with $B$ bones or joints. The set of all possible poses of this character is the power Lie group $SO(3)^B = SO(3) \times \cdots \times SO(3)$, which we call the pose Lie group (see Figure 3b).*

We give the interpretation of geodesics on $SO(3)^B$ in the context of animations of characters. One geodesic on $SO(3)$ is a curve $t \to \gamma(t)$ on $SO(3)$, *i.e.*, a sequence of rotations parameterized by a time $t$. In other words, a geodesic on $SO(3)$ represents a specific rotation motion for a given bone, and the velocity $\dot{\gamma}(t)$ at any time point is the tangent to the curve at that point. A geodesic on the pose Lie group $SO(3)^B$ represents a specific motion of the animated character.

We are interested in equipping this Lie group with a Riemannian metric to propose new interpolation methods. To this aim, we first show how to equip each component $SO(3)$ with a Riemannian metric. We will represent each inner-product $<,>_g$ at $T_g\mathcal{G}$ by its associated symmetric positive definite (SPD) matrix $Z_g$. We denote the matrix at the Lie algebra $T_eSO(3)$ as $Z_e = Z$. Formally, this inner product is defined as

$$<u, v>_e = u^T Z v, \tag{2}$$

for $u, v \in so(3) = T_eSO(3)$. The Lie group structure of $SO(3)$ makes it possible to define a family of Riemannian metrics on $SO(3)$.

## 4 METHODS

We propose a novel method to determine which choice of Riemannian metrics (defined in A.1) on the Pose Lie group $SO(3)^B$ yields better, more compressed, or more interpretable interpolation techniques.

### 4.1 OVERVIEW

**Problem statement:** Consider the animation of an articulated character that is made up of $B$ bones, and call it the *ground truth animation $A_G$*, see Figure 4 (left). We denote $F$ the number

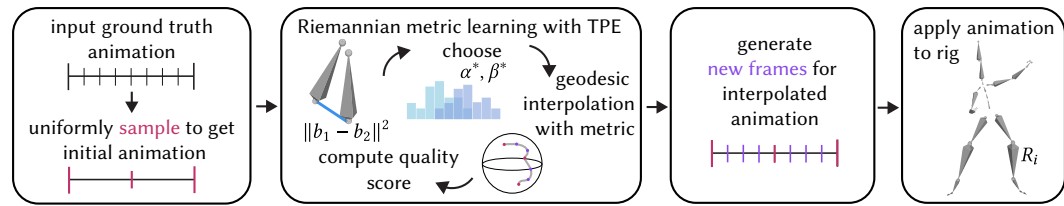

Figure 4: Pipeline of our method. Given an animation, we downsample it, and upsample new in-between frames using a search sweep for optimal parameters. Then we apply the animation to the rig and quantitatively and qualitatively analyse the results.

of frames in $A_G$. This animation is represented as a sequence of $F$ poses on the Pose Lie group $SO(3)^B$, so that:

$$A_G(t) \in SO(3)^B \qquad \text{for each time } t \in [t_1, t_F], \tag{3}$$

between the first and last frames $t_1$ and $t_F$ respectively.

Our goal is to learn the Riemannian metric $<,>$ on the Pose Lie group $SO(3)^B$ that best describes the animated character's motion in $A_G$, in the following sense: the animation $A_G$ can be downsampled (compressed) to a lower frame rate $F'$, such that the geodesic interpolation with metric $<,>$ brings it back to its original (higher) frame rate $F$ with the highest accuracy. Once $<,>$ is learned, it can be used for creative design in digital creation, including extracting perceptual insights on the character's motion in $A_G$, or for compression.

**Notations:** The metric $<,>$ is the result of an optimization problem that depends on $A_G$ and $F'$, for which we introduce notations. Consider a sampling rate $0 < s < 1$. We call *initial animation*, and denote it $A_I$, the animation obtained after uniformly downsampling the ground truth animation $A_G$ of frame rate $F$ to the lower frame rate $F' = sF$, see Figure 4 (pink). For example, if we have a ground truth animation of $F = 60$ frames, and a sampling rate of $s = 0.2$, the initial animation will have $F' = 12$ frames, each 5 frames apart in the ground truth. We call *interpolated* or *upsampled animation*, and denote it $A_U$, the animation obtained by upsampling the initial animation $A_I$ back up to the ground truth frame rate $F$, see Figure 4 (purple). We note that $A_U$ depends on the interpolation technique used: in particular, in the case of a geodesic interpolation, $A_U$ depends on the choice of metric $<,>$.

Reformulated using these notations, our goal is to learn the metric $<,>$ so that $A_U$ is as close as possible to $A_G$, according to a quality score $Q$. Figure 4 shows our pipeline.

## 4.2 RIEMANNIAN METRIC LEARNING

We propose to learn the metric $<,>$ that most accurately describes the motion of a given animated character. We restrict our optimization to a set of invariant Riemannian metrics on $SO(3)^B$, which provides a convenient parameterization of $<,>$. The definition and practical implementation of invariant Riemannian metrics is detailed in Appendix A.

**Metric Parameterization** Consider one $SO(3)$ within the power Lie group $SO(3)^B$. We can parameterize a Riemannian metric on the Lie group $SO(3)$ by an inner product matrix $Z$ on its Lie algebra (see Appendix A). The matrix $Z$ must be symmetric positive definite, meaning it can be decomposed into $Z = P^T D P$, where $D$ is a diagonal matrix whose values are strictly positive, and $P$ is orthogonal. We will restrict our investigation to specific matrices $Z$:

$$Z = \begin{bmatrix} 1 & 0 & 0 \\ 0 & \alpha & 0 \\ 0 & 0 & \beta \end{bmatrix}, \qquad \text{with } \alpha, \beta > 0, \tag{4}$$

on each component $SO(3)$ within the Pose Lie group $SO(3)^B$. In other words, we restrict ourselves to matrices $Z$ where the orthogonal component $P$ is taken to be the identity matrix and learn the optimal $\alpha, \beta$ values for each matrix $Z$ corresponding to each $SO(3)$ within the power $SO(3)^B$, in order to best reconstruct the animation. Our metric on $SO(3)^B$ is thus parameterized by the set of parameters: $\{\alpha_1, \beta_1, ..., \alpha_B, \beta_B\}$, written $\{\alpha, \beta\}$ for short. We also add a categorical parameter, called inv, which indicates whether whether we propagate the inner-product $Z$ with left or right translations: i.e., whether the resulting metric $<,>$ is left- or right- invariant (see Appendix A). This

parameterization does not cover every metric on $SO(3)^B$; yet, it encodes a $4B$-dimensional family of metrics where we can perform metric learning.

**Geodesic Interpolation** Consider a bone $b$ and two frames $i, j$ that are consecutive in the initial animation $A_I$ and $j - i + 1$ frames apart in the ground-truth animation $A_G$, i.e., $A_I(b, i) = A_G(b, i) = R_i \in SO(3)$ and $A_I(b, j) = A_G(b, j) = R_j \in SO(3)$. Given a metric $<,>$, we compute the geodesic $\gamma$ on $SO(3)$ such that $\gamma(0) = R_i$ and $\gamma(1) = R_j$ and the energy $E(\gamma)$ measured with $<,>$ is minimal according to Definition 3.2. The main challenge is to compute the initial tangent vector $u_0 = \dot{\gamma}(0)$ required to shoot from $\gamma(0)$ to $\gamma(1)$. This requires to numerically invert the Exp map defined in the previous section, i.e., solving the optimization problem:

$$u_0 = \underset{u \in T_{R_i} SO(3)}{\arg\min} \|\text{Exp}_{R_i}(u) - R_j\|^2. \tag{5}$$

The tangent vector $u_0$ then yields values of $A_U$ between frames $i$ and $j$ as: $A_U(b, t) = \text{Exp}_{R_i}(t.u_0)$ for $t \in [0, 1]$. We observe that we do not have a closed form expression for the interpolating geodesic, which is instead computed via numerical integration and optimization.

**Optimization Criteria: Quality Metrics** The upsampled animation $A_U$ is obtained by geodesic interpolation, which depends on the invariant Riemannian metric $<,>$ that is itself parameterized by $\alpha, \beta$ and `inv`. Thus, we write $A_U$ as a function of $\alpha, \beta, \text{inv}$: $A_U(\alpha, \beta, \text{inv})$. We detail here how we find the optimal parameters $\alpha, \beta, \text{inv}$ and thus the optimal Riemannian metric $<,>$ for digital animations, see Figure 4 (center). Consider a quality metric $Q$ that denotes how close the interpolated animation $A_U(\alpha, \beta, \text{inv})$ is from the ground truth animation $A_G$. We get:

$$\alpha^*, \beta^*, \text{inv}^* = \underset{\alpha, \beta, \text{inv}}{\arg\min} Q\left(A_U(\alpha, \beta, \text{inv}), A_G\right), \tag{6}$$

for $\alpha, \beta \in (\mathbb{R}_+^*)^B$ and `inv` in $\{\texttt{left}, \texttt{right}\}$. We will experiment with various quality metrics $Q$ within this optimization criterion. Our first quality metric quantifies the difference in position between two bones' endpoints:

$$Q_{loc}(b_1, b_2) = \|b_1 - b_2\|^2, \tag{7}$$

where $b_1$ and $b_2$ are the endpoint position of bones 1 and 2.

Our second quality metric quantifies the angle difference in rotation between two bones:

$$Q_{rot}(b_1, b_2) = \arccos\left[\frac{tr(b_1 b_2^T) - 1}{2}\right], \tag{8}$$

where in this case $b_1$ and $b_2$ are the rotation matrices of bones 1 and 2 respectively, see Figure 4 (purple). Our third quality metric $Q_{hyb}$ is a weighted sum of $Q_{loc}(b_1, b_2)$ and $Q_{rot}(b_1, b_2)$. Each of these three quality metrics is defined for a given bone of the rig, at a given frame. To get the quality scores $Q$ across bones and frames, we sum across the bones $b = 1, \dots, B$ with or without a weight $w_b > 0$ corresponding to the depth of that bone in the rig, and we average over all frames in the ground truth animation Wang et al. (2021). Thus, the total quality metric between a pose in the ground truth animation $A_G$ and the upsampled animation $A_U$ is:

$$Q = \frac{1}{F} \sum_{t=1}^{F} \sum_{b=1}^{B} w_b \tilde{Q}(A_U(b, t), A_G(b, t)), \tag{9}$$

and $\tilde{Q}$ equal to $Q_{loc}$, $Q_{rot}$ or $Q_{hyb}$. The dependency on $\alpha, \beta, \text{inv}$ is within the bone $b_{t,i}^U$ of the upsampled animation $A_U$.

**Optimization Method: Gradient-Free** We introduce the optimization method chosen to minimize the criterion of Eq. 6 and learn $\alpha^*$, $\beta^*$ and `inv` $^*$. This criterion does not have a closed form as a function of $\alpha, \beta$ and `inv`. Thus, we cannot compute its gradient, nor leverage any gradient-based optimization methods. Consequently, we propose to rely on a gradient-free optimization methods: the Tree-Structured Parzen Estimator (TPE). Tree-Structured Parzen Estimator algorithm Bergstra et al. (2011) is designed to find parameters that optimize a given criterion whose gradient is not available. TPE is an iterative process that uses history of evaluated parameters $\alpha, \beta, \text{inv}$ to create a probabilistic model, which is used to suggest the next set of parameters $\alpha, \beta, \text{inv}$ to evaluate, until the optimal set $\alpha^*, \beta^*, \text{inv}^*$ is reached. More details can be found in Appendix B.

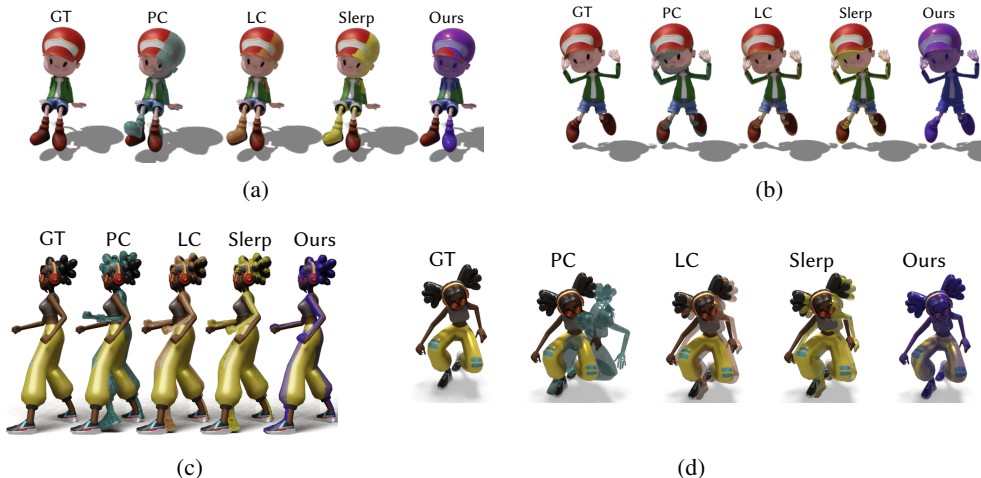

Figure 5: Figure 5a shows a boy sits and swings his legs back and forth. Frame 24 of the `Sitting` animation shows our geodesic almost perfectly recreating the pose. Figure 5b shows the entire purple overlay for our geodesic interpolation which indicates a high quality reconstruction. Figure 5c shows extremities like hands are captured more accurately in our method. Figure 5d, we capture the fast `Rolling` motion in frame 44.

**Implementation**  Our *ground truth animations* are downloaded motion capture sequences from Adobe Mixamo at 30 frames per second Adobe (2023). All animations are imported to Blender, which we use to visualize, render, and export animation data Community (2023). File sizes are computed as the sum of sizes (in bytes) of exported bone locations and rotations to NumPy files Harris et al. (2020). The Riemannian metric learning with TPE is performed using `HyperOpt` Bergstra et al. (2013), `Tune` Liaw et al. (2018) and `Wandb` to log the results Biewald (2020).

For cartesian linear interpolation, we linearly interpolate the locations as well as the rotations in the form of component-wise quaternion interpolation. Blender's quaternion interpolation was once implemented this way but was problematic since it can yield invalid (non-unit) quaternions. Blender has since updated to using a version of spherical linear interpolation (slerp), which we also compare to.

During geodesic interpolation on $SO(3)$, we generate new rotation matrices representing the orientation of each bone at a frame using the implementation of invariant Riemannian metrics parameterized by $\alpha, \beta,$ `inv` and available through the Geomstats library Miolane et al. (2020). In order to compute the quality metrics, we need to recover the new bone positions $b$ at each frame given orientations $R \in SO(3)$ and root bone position. To do so, we start from the root bone of the rig (*e.g.* hips) and traverse the tree breadth first, applying each new rotation to the bones on that "level" of the tree, computing the new positions, iteratively until we have leaf node (*e.g.* fingertips) positions.

## 5  RESULTS

We compare our geodesic interpolations (purple) to the three most commonly used schemes: piecewise constant (PC, teal), linear cartesian (LC, orange), spherical linear (slerp, yellow), on 5 different increasingly complex Mixamo animations: `Pitching`, `Rolling`, `Punching`, `Jumping`, and `Sitting`. Our supplemental video contains the full animations.

**Perceptual Accuracy**  We visualize and qualitatively compare the accuracy of each interpolation scheme. We present this comparison using a sampling rate of $s = 0.3$ in Figs. 5a-5d, while corresponding figures for other sampling rates can be found in the supplemental materials. Our visualizations show the ground truth animation, with the interpolation methods layered transparently over to highlight where the interpolation deviates from the original.

The `Pitching` in Fig. 1 has 24 bones and shows our method working with animations with a fixed root node. `Sitting` in Fig. 5a is an example where the fixed node is in the middle of the armature. `Jumping` contains vertical motion and rotations in the legs that are far apart, *i.e.* differ by a large angle close to $pi$. `Punching` animation in Fig. 5c shows horizontal translations with

contacts. For example, it would be undesirable for an interpolation to miss frames where her feet touch the floor to create an illusion of floating. Our approach outperforms traditional techniques as it most accurately interpolates characters within this diversity of animations: displaying a larger purple overlay in Figs. 5a-5b, effectively capturing extremities (hands and feet) in Fig. 5c as well as fast motions in Fig. 5d.

The `Rolling` animation is difficult because it has the complexity of all previous animations. The rotations of bones are large and flip upside down (see Fig. 5d). In this difficult setting, visual inspection shows that our interpolation performs particularly well.

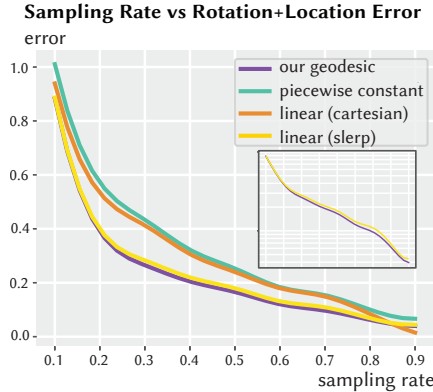

Figure 6: As the sampling rate for the `Pitching` animation increases—which means a higher number of frames in the initial animation—the error metric $Q_{hyb}$ decreases.

**Quantitative Accuracy and Compression** In addition to these perceptual comparison, we compare the interpolations' accuracies using the weighted error $Q_{hyb} = 0.5\ Q_{loc} + 0.5\ Q_{rot}$ and present it in Fig. 6 for the `Rolling` animation. The supplementary materials show these plots for the 4 other animations. Our approach presents the lowest error just in front of slerp's. Despite the seemingly small quantitative difference between these two, we note that Fig. 5d shows significant perceptually differences. Fig. 6 also allows us to evaluate our method in terms of compression: we require a lower sampling rate $s$ to achieve a given interpolation error (or accuracy). Consequently, this method can decrease the memory required to store animations: our compressed animation is a factor of $s$ smaller than the ground truth, plus the $B\alpha$ and $B\beta$ float values. The supplemental materials provide additional details on compression and exact file size.

## CONCLUSION AND FUTURE WORK

We presented a method for animation interpolation using geodesics on Riemannian manifolds where we learn the optimal metric. To our knowledge, this is the first time that Riemannian metric learning is proposed for computer graphics. We hope that these ideas will inspire other applications in this field. We showed that our method interpolates animations with high accuracy (both perceptually and quantitatively) on a variety of different motion capture sequences. Because we are able to accurately represent a high frame rate animation with very few frames, we achieve a compression rate that requires digital animators to pose fewer keyframes during the creation process.

Future work will perform further analyses of the metric parameters to reveal additional meaning and novel semantic intuition behind the motion. We will eventually provide animators full control over the parameters $\alpha$ and $\beta$ to change the interpolation style and foster a more interactive exploration. We will do so by integrating our family of geodesic interpolation into animation software and enable animators to play with different geodesics in real time. We are also interested in performing perceptual studies to investigate which interpolations animators and viewers prefer.

One can also explore how choice of keyframes impacts interpolation and compression results. Our experiments uniformly downsample the ground-truth animation. Yet, with an extremely low sampling rate, the downsampled animation consists of very few frames which might not capture all important actions. One can explore how a smart downsampling of the animation improves interpolation quality by ensuring that the most important frames are kept.

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

## A   APPENDIX: INVARIANT METRICS

**Definition A.1 (Invariant Riemannian metric)** *A left-invariant metric on a Lie group* $(G, \cdot)$ *is a Riemannian metric* $<, >$*, such that for all* $g, h \in G$ *and for all* $u, v \in T_g G$*, we have:*

$$\langle DL_h(g).u, DL_h(g).v >_{L_{hg}} = \langle u, v \rangle_g, \tag{10}$$

*where* $L_h$ *is the left translation by* $h$*, i.e.* $L_h g = h \cdot g$*, and* $DL_h$ *denotes its differential. In other words, the left translations are isometries for this metric. Similarly, we can define right-invariant metrics using the right translation* $R_h$*.*

We note that any Lie group $G$ admits a family of left (and right) invariant metrics, which we can obtain with a construction that we illustrate with $G = SO(3)$ in what follows. First, we define an inner-product on $so(3) = T_e SO(3)$ by providing a SPD matrix $Z$ as in Eq. 2. Then, this inner-product is turned into a Riemannian metric by propagating it to get an inner-product at each tangent space $T_h G$ using the differential of left-translations $DL_h(e)$ (resp. $DR_h(e)$) that we compute with automatic differentiation. We refer the reader to Guigui et al. (2022) for details. In practice, it means that the specification of a single matrix $Z$ provides us with a full Riemannian metric over $SO(3)$, which becomes a Riemannian manifold.

*Remark:* Intuitively, left-invariant metrics are metrics that are invariant with respect to change in inertial frame, but not with respect to change in body-fixed frame Žefran et al. (1996). Similarly, right-invariant metrics are invariant with respect to body-fixed frame but not with respect to changes in inertial frames.

**Proposition A.1 (Invariant Geodesic Equations Guigui & Pennec (2021))** *Consider    a    left-invariant metric* $<, >$ *on a Lie group* $G$ *with Lie algebra* $\mathfrak{g}$*. Its geodesics* $\gamma$ *verify the ordinary differential equation:*

$$\begin{cases} \dot{\gamma}(t) = dL_{\gamma(t)}\omega(t) \\ \dot{\omega}(t) = \mathrm{ad}^*_{\omega(t)}\omega(t) \end{cases}, \tag{11}$$

*where* $\mathrm{ad}^*$ *is the metric dual to the adjoint map, which is defined as:* $\forall a, b, c \in \mathfrak{g}, \langle \mathrm{ad}^*_a(b), c \rangle = \langle [a, c], b \rangle$*, where* $[,]$ *is the Lie algebra bracket.*

In the proposed geodesic interpolation method, our geodesics will be numerically computed by integrating ODE Eq. 11 with an Euler integration scheme. Given initial conditions $p = \gamma(0), u = \dot{\gamma}(0)$ for ODE Eq. 11, we denote $\mathrm{Exp}_p(u)$ the solution of ODE Eq. 11 at time $t = 1$, which is called the Riemannian exponential map associated with metric $<, >$. We note that the Riemannian metric $<, >$ is bi-invariant if and only if the adjoint map is skew-symmetric, i.e. $\mathrm{ad}^*_\omega(\omega) = 0$, which is what is used in the spherical interpolation of quaternions in the literature.

## B    APPENDIX: TPE ALGORITHM

The TPE algorithm consists of three steps. First, we start with a random selection of tuples $\alpha, \beta, \texttt{inv}$ and evaluate the criterion $Q$ corresponding to each tuple, i.e., the accuracy of the interpolated animation compared to the ground truth animation for each tuple given by Eq. 6. Second, we sort the collected criteria and divide them into two groups based on a quantile chosen in advance. The first group contains the tuples $\alpha, \beta, \texttt{inv}$ that gave the best (lowest) criteria $Q$ and the second group contains the other tuples with the worst (highest) criteria $Q$. We compute the two probability densities $p_1$, $p_2$ on the tuples' distributions in the first and second group respectively, using Parzen Estimators, *i.e.*, kernel density estimators. Finally, we draw new tuples from the probability density corresponding to the best group: $p_1$. We evaluate the new tuples in terms of an improvement score $I = p_1(\alpha, \beta, \texttt{inv})/p_2(\alpha, \beta, \texttt{inv})$ and return a set of new tuples that yields minimum values for $I$. Intuitively, we expect this set to correspond to the greatest expected improvement $I$ in the criterion $Q$. These new tuples are then evaluated on the criterion $Q$, and the steps are iterated. We continue the process until a convergence threshold is reached.

