# OpenReview forum: "Learning Riemannian Metrics for Interpolating Animations"
_ICLR.cc/2024/Conference — ICLR 2024 Conference Withdrawn Submission_

### Official Review · Reviewer_QSvS · 2023-10-22

**Soundness:** 3 good
**Presentation:** 2 fair
**Contribution:** 3 good
**Rating:** 6
**Confidence:** 4

**Summary:**

This paper has presented an interesting and inspiring study. It presented a method for animation interpolation using geodesics on Riemannian manifolds. This is the first time that Riemannian metric learning has been proposed for computer graphics. This method has shown very good qualitative results and good quantitative results.

**Strengths:**

The method is interesting, I like the math behind it. The results especially the video look quite nice. The amount of parameters need to be learnt is limited thus it should be much easier to interpret.

**Weaknesses:**

I didn't see the code, so I am not able to fully understand the full math behind it and how they actually implemented it.
I am not sure why the quantitative results are not that outstanding. From the videos, the proposed new method seems to be a much better method or kind of revolutionary method, however, the quantitative results do not show this when compared with the Slerp method. And to my knowledge Slerp is based on the special Riemannian manifold of a sphere on which the SO(3) naturally lay. Thus the metric used in Slerp is naturally based on the angles, it will be much more obvious the effect about the Riemannian metric learning, if the learnt metrics are written out and the difference between them compared to the one used in Slerp is discussed. So maybe we can see why the quantitative results are so similar. If the learnt metric is same or very similar to the Slerp one then maybe it is not that useful.

**Questions:**

1. I think more explanations and analysis of why the criterion of Eq. 6 is not a closed form is preferred in the appendix. Since I didn't see the code, in my naive thought, it is just matrices multiplications, thus we shall be able to use gradient optimization. Or actually, the computation of the initial tangent vector u0 = γ ̇(0) is involved in the whole optimization process, so it is not possible to use gradient-based optimization?

2. I would really prefer a runnable code with all the environment and data needed to fully understand the paper. With them, I think maybe this paper can have even higher scores.

3. The quantitative analysis is not sound, it might be rooted in the way of the quantitative measurement. I suggest try other measurements as well, since the qualitative results seem to be quite good.

4. I want to see the effect about the Riemannian metric learning by comparing the one used in Slerp and the learnt ones.

---

> ### Comment · Reviewer_QSvS · 2023-11-22
>
> If the author keeps ignoring my questions, I will definitely suspect the academic spirit of the author. A positive and encouraging comment does not mean the questions related don't need to be addressed.

---

> > ### Author Response · Authors · 2023-11-22
> >
> > Dear Reviewer QSvS,
> >
> > I am sorry I didn't write this comment earlier. I truly thank you for reading our paper and I appreciate the comments and questions. We have decided to withdraw the paper after all.
> >
> > But in response to your questions:
> > 1. Yes we need more explanation about our optimization choices. Thanks to you and Reviewer 3Sna, we will investigate further about a closed-form solution wrt to alpha and beta and consider gradient-based optimization if applicable.
> > 2. We could definitely publish/submit the code in another submission. Great suggestion.
> > 3. Do you mean to use other metrics in our optimization or try to measure our output's success with other measurements? It's true that qualitatively our results look much better than our quantitative analysis.
> > 4.
> > >  if the learnt metrics are written out and the difference between them compared to the one used in Slerp is discussed. So maybe we can see why the quantitative results are so similar.
> >
> > How do you mean to compare the learned metrics to slerp? You are correct that Slerp is the equivalent of geodesics on SO(3) with the canonical metric. Can you clarify what you mean by "learnt metrics .. written out"? It would be great for us to compare the metrics better so I am curious what you mean by writing the metrics out.
> >
> > Thanks,
> > Authors

---

### Official Review · Reviewer_3Sna · 2023-10-28

**Soundness:** 1 poor
**Presentation:** 1 poor
**Contribution:** 2 fair
**Rating:** 1
**Confidence:** 5

**Summary:**

The authors present a new technique of SO(3) interpolation parameterized by a family of Riemannian metrics. The method learns the Riemannian metric that best describes a given ground truth animation, which is a sequence of SO(3) associated with a character's rig. The metric has three parameters, α and β for the diagonal matrices used for the metric, and a categorical parameter "inv" for invariance.
The authors adopt the Tree-Structured Parzen Estimator as a gradient-free optimization method.
The authors use a public data from Mixamo animations to compare the method with the piecewise constant, Cartesian linear interpolation, and spherical linear interpolation (slerp).

**Strengths:**

- The authors present a simple geodesic of interpolation method that can be addressed as a useful extension of Slerp.
- The mathematics behind the proposed method is well explained.

**Weaknesses:**

The second contribution is not well presented. The paper significantly lacks experimental analysis to support the effect of α and β. The guideline is also unclear.
As far as I understand, the proposed method requires the ground truth animation to learn the parameters, so the method is usable for compression but not for creative design. Even as a compression method, there is no experiment and discussion about the relationship between compression rate and accuracy degradation.
The authors raised two important points as future work: further analysis of the metric parameters and full control over the parameters. These points are definitely crucial and necessary for acceptance.
According to the above issues, section 2 about related work is not appropriate in its current form.
Since many methods are not appropriately addressed, some of them should still be comparable in the given problem setting.


Detailed comments.
- There is an unnecessary term in the caption of Figure 3.
- "whether whether" at the bottom of page 6.
- The authors introduced "weight" in equation 9, but did not evaluate it.
- The authors simply took a weighted metric $Q_{hyb}$ as 0.5 $Q_{loc}$ + 0.5 $Q_{rot}$, but the former depends on the scaling of the scene and the latter is scale-invariant. The current result should also depend on the scale of the scene, which is inappropriate.
- Although the authors show a quantitative result, it seems inappropriate. The authors' statement means that the quantitative result does not reflect the quality: "Despite the seemingly small quantitative difference between these two, we note that Fig. 5d shows significant perceptually differences."
- The statement lacks support: "we are able to accurately represent a high frame rate animation with very few frames, we achieve a compression rate that requires digital animators to pose fewer keyframes during".

**Questions:**

The reviewer also has a question about the third contribution. The authors claim the need for gradient-free optimization as "This criterion does not have a closed form as a function of α, β and inv. Thus, we cannot compute its gradient, nor leverage any gradient-based optimization methods". I'm not confident since the formulation about A_{U}( α, β, inv ) is not clearly given, however, in the reviewers' understanding, the formulation is differentiable with α, β (using autograd techniques). Please clarify how it is not differentiable. As for inv, it would clearly be indifferentiable, but just solving both cases and choosing the better one is enough.

---

> ### Author Response · Authors · 2023-11-20
> **Question about gradient-free optimization**
>
> Dear Reviewer 3Sna,
>
> Thank you for the careful read of our paper.
>
> There is no closed-form for the geodesics with alpha, beta thus they need to be computed numerically by integrating the geodesic equation. Do you mean that we can differentiate through this integration with automatic differentiation tools?
>
> When you say A_{U}(α, β) is differentiable wrt α, β do you mean that there is a closed-form or that a gradient exists so we could use another algorithm than the gradient-free option we chose?
>
> Thanks,
> Authors

---

> ### Comment · Reviewer_3Sna · 2023-11-21
>
> Is it different from the geodesic computed by Log/Exp for the anisotropically scaled ($Z$=diag(1,α, β)) matrix of rotation, which is depicted in Fig.3(b)?

---

> > ### Author Response · Authors · 2023-11-21
> >
> > No you're correct, we are solving for the geodesic for the anisotropically scaled rotation in fig 3b. This geodesic is differentiable?

---

> > > ### Comment · Reviewer_3Sna · 2023-11-21
> > >
> > > If so, I think it does have closed form and differentiable using matrix Exp and Log.
> > > Although Pytoch does not provide matrix_log, an implementation is found in matlab.
> > > https://www.mathworks.com/matlabcentral/fileexchange/38894-matrix-logarithm-with-frechet-derivatives-and-condition-number

---

### Official Review · Reviewer_T2GP · 2023-10-29

**Soundness:** 1 poor
**Presentation:** 3 good
**Contribution:** 1 poor
**Rating:** 3
**Confidence:** 4

**Summary:**

The paper proposes to learn a metric to interpolate poses represented by the tree structure of joint rotations as used in computer animation. Specifically, the metric is defined as the product of the geodesics over the scaled Riemannian manifold of individual joint rotations. The authors propose to use TPE to optimize the scaling parameters (alpha and beta per joint) and the left and right invariance. The paper shows the results of the pose interpolations compared to other basic interpolation strategies (piecewise constant, linear cartesian, and spherical interpolation).

**Strengths:**

The metric learning of the poses in the skeletal structures as used in the digital humans and character representations is an important problem.

**Weaknesses:**

The interpolation of poses defined by the tree of SO(3)s using the Riemannian manifold is a well-studied topic, especially in computer graphics. "Practical Parameterization of Rotations Using the Exponential Map" [Grassia 1998] is one such paper that became the foundation of the pose representation for many papers. For example, "Geostatistical Motion Interpolation" [Mukai and Kuriyama 2005], which the authors have cited but simply disregard because it is not implemented in the animation software, uses the concatenation of the exponential maps of the joint rotations as the pose representation. "SMPL: A Skinned Multi-Person Linear Model" [Loper et al. 2015] also uses the rotation vector (the axis vector divided by the angle component in the axis-angle representation) per joint, which is equivalent to the exponential map (the Riemannian manifold) of rotations. I wish the authors had paid attention to how these previous papers treated this problem in more depth.

More specifically, the authors' proposal of collapsing the pose representation by taking the product of the joint rotations (pose Lie group in definition 3.4) does not make sense. Intuitively, if this is a single chain of joints, the product is the rotation of the last joint relative to the coordinate frame of the first joint. Even if we assume the single chain structure, the product represents only the last joint's rotation relative to some coordinate frame and is not representative of the full pose. Even worse, because the joints are in the tree structure, taking the product of all joint rotations does not make sense at all (e.g., the left foot is not chained from the left wrist.)

Therefore, it makes much more sense to treat the poses in high-dimensional space of the concatenation of the joint transform representations (e.g. exponential maps) and optimize the interpolation parameters in this space, which is essentially [Mukai and Kuriyama 2005] and many data-driven motion papers do. The previous approaches, e.g. [Mukai and Kuriyama 2005], can be seen as learning the scaling parameters of this paper (alphas and betas) but keeping the individual components.

If authors argue that taking the product of all joint rotations makes sense, this has to be shown in numbers and visuals, compared to methods such as [Mukai and Kuriyama 2005] with more examples.

**Questions:**

N/A

---

> ### Author Response · Authors · 2023-11-20
> **Confusion about product of joint rotations**
>
> Dear Reviewer T2GP,
>
> Thank you for taking the time to review our paper.
>
> There seems to be a misunderstanding. We do not mean the matrix product of the individual rotation matrices as: R1….RN but rather the product in the sense of product spaces which gives (R1, …, RN) which is indeed representative of the full pose.
> “high-dimensional space of the concatenation of the joint transform representations”
> --> This is what we do, the product of the spaces SO(3) x …. x SO(3) precisely means concatenation. We do take the exponential map according to the riemannian metric.
>
> Was this indeed your interpretation of the paper?
>
> Thanks,
> Authors

---

> > ### Comment · Reviewer_T2GP · 2023-11-20
> >
> > > the product of the spaces SO(3) x …. x SO(3) precisely means concatenation
> >
> > Whatever the SO(3) representation is (matrix, quaternion, or whatever), a usual convention for a product of rotations R_2 x R_1 is applying the first rotation R_1, then the second rotation R_2 to end up with the final rotation. If the authors wish to define that the product here is a concatenation, then this has to be clarified in the paper. I would suggest using different notations, though.
> >
> > To clarify, are the authors saying that a pose in the pose Lie group SO(3)^B is meant to have the dimension of B times the dimension of the individual rotation representation?
> >
> > Either way, the paper must be compared against other methods using the exponential map of the rotations, not just against trivial interpolation strategies like Slerp.

---

> > > ### Author Response · Authors · 2023-11-21
> > >
> > > > To clarify, are the authors saying that a pose in the pose Lie group SO(3)^B is meant to have the dimension of B times the dimension of the individual rotation representation?
> > >
> > > Yes, that's correct. A pose in our space is the concatenation of B rotations (in our case, quaternions), where B is the number of bones in the rig.

---

### Official Review · Reviewer_8Wt5 · 2023-10-31

**Soundness:** 3 good
**Presentation:** 3 good
**Contribution:** 2 fair
**Rating:** 5
**Confidence:** 4

**Summary:**

This paper proposes a method for interpolating between character animation poses (i.e. keyframes).  Given a ground truth animation sequence, they propose to optimize for a Riemannian metric on the space of poses (the Lie group $SO(3)^B$), which minimizes a quality metric between the ground truth animation sequence, and one that has been first downsampled then upsampled by interpolating between keyframes along a geodesic path on a manifold equipped with the metric.  They provide comparisons between their interpolation scheme, and other well-known methods, and assert that their interpolation scheme can also be applied toward animation sequence compression.

**Strengths:**

The paper is well-written and offers strong exposition containing relevant mathematical background.  Building off of that groundwork, the ideas presented in the paper are clear and well-formulated.  The established problem space of animation interpolation is interesting (of particular interest to a computer graphics audience).  Their proposed method is concise, and addresses a targeted, clearly-identified problem.

**Weaknesses:**

The paper claims that their method may provide a "novel diversity of inbetweening methods for digital creators", yet it is not clear to me that their method has actually come close to achieving this.  No user studies are provided to support the claim, and no comparison _within_ their method between different choices of metrics is provided to illustrate how different metrics may result in different creative outcomes.  Further, I am skeptical that selecting a particular Riemannian metric would be an ideal interface for an artist to use in their workflow.

Evaluation is fairly limited, showing results on only a small set of motions.  Qualitative results are limited and not validated for statistical significance (e.g. via a user study).  I personally find that overlaying the original animation on top of the upsampled animation makes it difficult to appreciate the quality of the result.  Also, quantitative results (e.g. those shown in Figure 6 in the main paper) suggest that their method is only incrementally preferable to the leading alternative (slerp).

Though I appreciate that the authors convey that their proposed approach is compatible with a variety of quality metrics, and that they have presented multiple such metrics, more comparison between the metrics would be helpful, so that the reader may know the authors' decisive/data-driven opinion on which metric is preferable.

**Questions:**

Regarding the optimization method chosen for this particular problem, are there any limitations, such as defining the search space for initial values of the parameters?

It seems that the learned metric would be dependent on the specific character rig as well as which set of animations are included in the optimization.  Could you comment on how this learned metric would generalize to other rigs and/or animations?

Given the memory requirements to store character animation data, it's not clear to me that compression is a compelling motivation.  That is to say, keyframe animation data seems to be on a MUCH smaller scale than other forms of data e.g. video or web-search indexes.  Could you elaborate upon whether compression of keyframe animation data is an established problem for real-world practitioners?

---

> ### Author Response · Authors · 2023-11-22
>
> Dear Reviewer 8Wt5,
>
> Thank you for reviewing our paper. We appreciate the comments, questions, and criticisms. We will be withdrawing this paper after all but your insight is greatly appreciated.
>
> Regarding your questions:
> > Regarding the optimization method chosen for this particular problem, are there any limitations, such as defining the search space for initial values of the parameters?
>
> Yes, in our optimization, we define the search space as a range of alpha, beta values between 0.0001 and 15. We chose these parameters because experimentally we found that the integration of the Log is numerically unstable for values outside that range. We do not choose initial values explicitly but perhaps the library we use has default values.
>
> > It seems that the learned metric would be dependent on the specific character rig as well as which set of animations are included in the optimization. Could you comment on how this learned metric would generalize to other rigs and/or animations?
>
> Yes you are correct that the learned metric is specific to both the rig and the animation. In our examples, some of the characters have different rigs. They have mostly the same structure but a different number of bones. We observed that our metric learning seems to be successful on each rig and animation we had. We tried and failed to find animated rigs of non-humanoid characters, which sometimes have bones connected in different ways. This would be a goal of ours to test our code on a larger variety of rigs, although we expect the quality of results to be similar to what we've shown.
>
> > Given the memory requirements to store character animation data, it's not clear to me that compression is a compelling motivation. That is to say, keyframe animation data seems to be on a MUCH smaller scale than other forms of data e.g. video or web-search indexes. Could you elaborate upon whether compression of keyframe animation data is an established problem for real-world practitioners?
>
> I do believe compression of keyframe animation is a useful problem to solve. For example, in video game engines, animations of characters and their poses need to be stored in memory and accessed in real time during gameplay. You are correct that it is on a smaller scale than other data formats but it is still important.
>
> Thanks,
> Authors

---

> > ### Comment · Reviewer_8Wt5 · 2023-11-22
> >
> > Dear authors, thank you for clarifying my questions, I appreciate your thoughtful responses.
> > As of now, I maintain my original score.  Wishing you well as you continue to strengthen this work.

---

### Author Response · Authors · 2023-11-22

We thank all the reviewers for their very valuable insights and comments. Based on their excellent suggestions, we will work towards adding more experiments and better rephrase our contributions.
We will withdraw the submission to make time and prepare a major revision. We hope to re-submit in the near future a much improved version of this work to another venue.